# DYSTIL: Dynamic Strategy Induction with Large Language Models for Reinforcement Learning

## Abstract

Reinforcement learning from expert demonstrations has long remained a challenging research problem, and existing methods resorting to behavioral cloning plus further RL training often suffer from poor generalization, low sample efficiency, and poor model interpretability. Inspired by the strong reasoning abilities of large language models (LLMs), we propose a novel strategy-based neuro-symbolic reinforcement learning framework integrated with LLMs called Dynamic Strategy Induction with Llms for reinforcement learning (DYSTIL) to overcome these limitations. DYSTIL dynamically queries a strategy-generating LLM to induce textual strategies based on advantage estimations and expert demonstrations, and gradually internalizes induced strategies into the RL agent through policy optimization to improve its performance through boosting policy generalization and enhancing sample efficiency. It also provides a direct textual channel to observe and interpret the evolution of the policy's underlying strategies during training. We test DYSTIL over challenging RL environments from Minigrid and BabyAI, and empirically demonstrate that DYSTIL significantly outperforms state-of-the-art baseline methods by 17.75% success rate on average while also enjoying higher sample efficiency during the learning process.

## 1 Introduction

Many important, yet challenging, reinforcement learning tasks (Chevalier-Boisvert et al., 2023; 2019; Mnih et al., 2013) are highly hierarchical and structural, have sparse and delayed rewards, and require complex reasoning procedures based on understanding of higher-level abstractions. In practice, classical reinforcement learning algorithms often fail to learn these difficult RL tasks well from scratch, because of the difficulty in collecting meaningful reward signals during exploration and the lack of support for higher-level abstraction and reasoning. Therefore, it is often necessary to collect a set of expert demonstration trajectories to aid reinforcement learning over these tasks (Ramírez et al., 2022). Most of the best existing methods for reinforcement learning from expert demonstrations typically first employ behavioral cloning (Pomerleau, 1988) to train the RL agent's policy generator to imitate the behavior and action decisions of the expert through supervised learning. They then feed the agent into a more advanced RL algorithm (such as Proximal Policy Optimization (Schulman et al., 2017)) to further improve its performance.

This approach of behavioral cloning plus further RL training suffers from several severe limitations: (1) expert demonstrations are often expensive or hard to collect, so typically the amount of expert demonstration trajectories is quite limited; (2) these limited expert demonstrations usually can only cover a small region of the state space, and thus behavioral cloning over them often tends to cause overfitting and results in poor generalization of the learned policy; (3) this approach can not enable the RL agent to acquire higher-level abstractions and understanding of the RL tasks, thus limiting the efficiency with which it utilizes training samples as well as the level of performance it can achieve; (4) this approach treats the policy network of the RL agent as a black box and thus suffers from low model transparency and interpretability.

To overcome the aforementioned limitations, in this paper we present DYSTIL, a novel strategy-based neuro-symbolic reinforcement learning framework integrated with LLMs called Dynamic

STRATEGY INDUCTION WITH LLMS FOR REINFORCEMENT LEARNING FROM EXPERT DEMONSTRATIONS (DYSTIL). In our daily lives, we can often observe this interesting phenomenon: when a teacher tries to teach a skill to a student, the most effective and efficient teaching method often involves more than merely asking the student to memorize all the details of specific actions. It is usually also complemented by clear explanation of the general strategies, principles, and ways of thinking for correctly approaching new scenarios when applying this skill. Inspired by this key observation and the strong abilities of knowledge induction (Zhu et al., 2024; Han et al., 2024) and reasoning (Wei et al., 2022; Pan et al., 2023) exhibited by state-of-the-art large language models (LLMs), we propose to leverage LLMs to help RL algorithms to induce generalizable strategies and learn higher-level abstractions about RL tasks from expert demonstrations, and we formulate our proposed new learning framework into DYSTIL. DYSTIL dynamically queries a large-scale LLM to induce textual strategies based on advantage estimations and expert demonstrations, and gradually internalizes induced strategies into the RL agent through policy optimization to improve its performance.

To empirically assess the effectiveness of DYSTIL, we run comprehensive experiments and ablation studies over four challenging RL environments from Minigrid (Chevalier-Boisvert et al., 2023) and BabyAI (Chevalier-Boisvert et al., 2019). Our experiment results show that DYSTIL achieves significantly superior learning performance and has higher sample efficiency over existing baseline methods across different RL environments. On average DYSTIL outperforms the strongest baseline method by 17.75% success rate across the four RL environments.

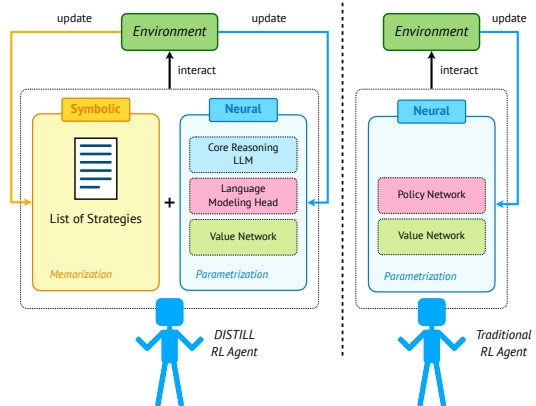

Figure 1: The neuro-symbolic nature of our DYSTIL RL agent.

To summarize, DYSTIL has the following key advantages and contributions: (1) it adopts a novel neuro-symbolic architecture for the RL agent to enable good synergy between higher-level strategy acquisition (the symbolic component) and parametrized policy optimization (the neural component); (2) it achieves effective knowledge distillation in the form of strategy induction from large-scale closed-source LLMs onto lightweight open-source LLMs to largely improve the generalizability of the agent's policy; (3) it achieves significantly better learning performance and sample efficiency over baseline methods during evaluation; (4) it largely enhances the model transparency and interpretability of the RL agent by providing a direct textual channel to observe and interpret the evolution of the policy's underlying strategies during RL training. Our work opens up new possibilities in leveraging LLMs to generate textual strategies to enhance the performance, efficiency and interpretability of reinforcement learning algorithms through a neuro-symbolic approach.

## 2 DYSTIL: DYNAMIC STRATEGY INDUCTION WITH LLMS FOR REINFORCEMENT LEARNING

### 2.1 PRELIMINARIES:

**Problem Formulation** This paper targets at the following *reinforcement learning from expert demonstration* problem, which can be formulated under the framework of partially-observable Markov decision processes (POMDPs) (Kaelbling et al., 1998): We have an agent $\mathcal{L}$ in a reinforcement learning environment $E$, which is a POMDP with observation space $\mathcal{O}$ and action space $\mathcal{A}$. Additionally, the agent $\mathcal{L}$ is provided with a set $\mathcal{D}$ of $N$ expert demonstration trajectories, where $\mathcal{D} = \{d_1, d_2, ..., d_N\}$. Each expert demonstration trajectory $d_i$ in $\mathcal{D}$ is a list of observation-action pairs in sequential order demonstrated by the expert in the environment $E$, where $d_i = [(o_1^{d_i}, a_1^{d_i}), (o_2^{d_i}, a_2^{d_i}), ..., (o_{T_{d_i}}^{d_i}, a_{T_{d_i}}^{d_i})]$. The goal of the agent $\mathcal{L}$ is to learn an optimal policy $\pi_{\mathcal{L}}$ that maximizes its expected total discounted reward $E[\sum_{t=0}^{\infty} \gamma^t r_t \mid \pi_{\mathcal{L}}]$, where $\gamma$ is the discount factor and $r_t$ is the reward that the agent receives at time step $t$.

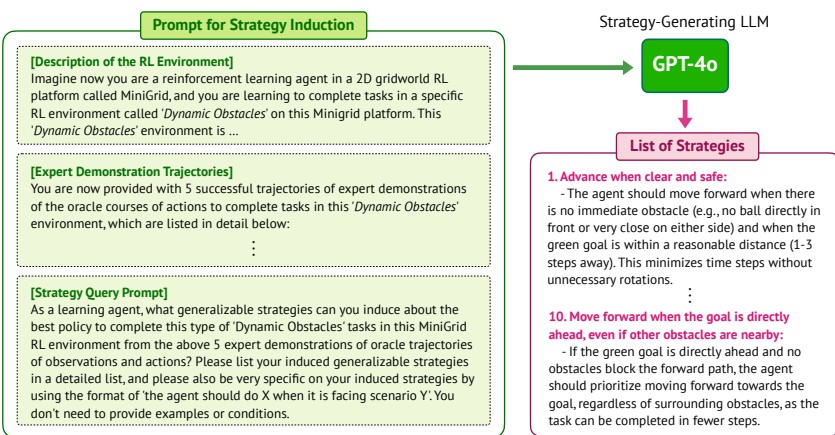

Figure 2: An example strategy induction process from expert demonstrations in GPT-4o (OpenAI, 2024) for the Dynamic Obstacles RL environment from the Minigrid library (Chevalier-Boisvert et al., 2023). See Appendix F for the complete list of strategies induced in this example.

**Language Grounding** In DYSTIL we take a language-grounded approach to reinforcement learning. Previous work (Carta et al., 2023) has demonstrated that running reinforcement learning using an LLM policy generator over textual descriptions of agent observations instead of the original raw observations can largely boost learning performance and sample efficiency. A crucial prerequisite for language-grounded RL is having access to a good observation-to-text converter that can convert the agent's raw observation information (such as images or state tensors) about the environment into rich and accurate textual descriptions in natural language. In general, such an observation-to-text converter can be either rule-based (such as BabyAI-text proposed in (Carta et al., 2023)) or trained with neural network architectures. Without loss of generality, in this work we assume that our RL agent has access to an accurate and well-functioning observation-to-text converter $\mathcal{C}_{o\to t}$, which is a safe assumption given the recent rapid advances in pre-trained multimodal foundation models (Li et al., 2024). Please see Figure 7 in Appendix C for a concrete example of observation-to-text transformation using BabyAI-text.

## 2.2 STRATEGY INDUCTION WITH LLMS FROM EXPERT DEMONSTRATIONS

Recent research works have demonstrated the ability of LLMs to automatically extract generalizable rules, knowledge and insight from examples (Zhu et al., 2024; Zhao et al., 2024). Inspired by these works, here we focus on using LLMs to automatically induce useful and generalizable strategies for completing tasks in reinforcement learning environments from trajectories of expert demonstrations.

We adapt and extend the prompting method in (Zhao et al., 2024) to design our prompt for automatic RL strategy induction. Our prompt has three components: (1) *Description of the RL Environment*; (2) *Expert Demonstration Trajectories*: this component includes a full textual description for each of the expert demonstration trajectories in $\mathcal{D}$ including its goal and a concatenation of the textual descriptions of all {observation, action} pairs in sequential order; and (3) *Strategy Query Prompt*: this paragragh describes our expectations for the kind of strategies that the LLM should induce from expert demonstrations and generate for us. Figure 2 demonstrates a concrete example of this strategy induction process from expert demonstrations in GPT-4o for an RL environment called Dynamic Obstacles from the Minigrid library (Chevalier-Boisvert et al., 2023). As we can see in Figure 2, the list of strategies induced by GPT-4o is indeed very relevant to successfully completing tasks in this Dynamic Obstacles RL environment, and also coincides with human intuition.

In the DYSTIL framework, we call the LLM used for inducing strategies the *strategy-generating LLM*, which is typically a SOTA large-scale LLM (e.g. GPT-4o (OpenAI, 2024)) that has strong reasoning abilities.

## 2.3 A NEW NEURO-SYMBOLIC MODEL ARCHITECTURE FOR DYSTIL RL AGENTS

In coordination with the DYSTIL learning framework, we design a novel strategy-based neuro-symbolic model architecture for our DYSTIL RL agent. Our new model architecture for DYSTIL RL agents is upgraded from the agent model architecture introduced in Carta et al. (2023) and augmented with strategies. This strategy augmentation transforms the original neural-only RL agent model in Carta et al. (2023) into a neuro-symbolic RL agent model. More specifically, our new model architecture has the following four components as illustrated in Figure 3.

**Input Concatenator** For each time step of decision making in an RL environment, the input to our DYS-TIL RL agent model is constructed by concatenating the following texts together: (1) a concise and essential *description of the environment*, such as the set of actions that an agent can take in the environment; (2) *goal* of the RL agent; (3) the list of *induced strategies* cur-

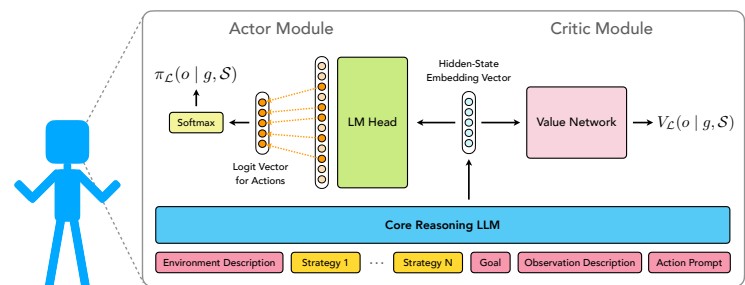

Figure 3: The strategy-based neuro-symbolic model architecture of our DYSTIL RL agents.

rently stored in the RL agent's memory; (4) a detailed textual description of the RL agent's *observation* of the 'state' of the environment at the current time step, which includes the agent's observation at the current time step and a history of $H$ (observation, action) pairs from the previous $H$ time steps in the agent's current trajectory; (5) an *action prompting prefix* (i.e. 'Action H:'). See Figure 6 in Appendix A for an example textual input into our new agent model following this template for $H = 2$.

**Core Reasoning LLM** The core information processing and reasoning module of our model is a lightweight open-source LLM for autoregressive language modeling that is open to efficient parameter tuning, such as Meta Llama 3.1 8B (Meta, 2024). We call this module the *core reasoning LLM* (in order to distinguish from the strategy-generating LLM introduced in Section 2.2). We directly feed the aforementioned dynamically-constructed textual input into this core reasoning LLM, and on its output side we take the last-layer hidden-state vector of the last token, which we denote as $w$.

**Actor Module** For the actor module of our agent model, we feed that hidden-state vector $w$ into the innate pre-trained language modeling head of the core reasoning LLM. From its output, we fetch the logit values for the first tokens of all action names and group them together into a shorter logit vector, and then apply the softmax function on it to obtain a probability distribution over all possible actions as our RL agent $\mathcal{L}$'s policy $\pi_{\mathcal{L}}(o \mid g, \mathcal{S})$.

**Critic Module** For the critic module of our agent model, we directly feed that hidden-state vector $w$ directly into a value network that project $w$ into a real number as the value of the value function $V_{\mathcal{L}}(o \mid g, \mathcal{S})$.

## 2.4 DYNAMIC STRATEGY INDUCTION WITH LLMS BASED ON PROXIMAL POLICY OPTIMIZATION

The induction method introduced in Section 2.2 is often able to generate a useful list of strategies that can help RL agents make better decisions in RL tasks, but it also has one prominent limitation: it is a one-time query and the induced list of strategies will remain static over time. As a result, if the initial one-time induced list of strategies from the strategy-generating LLM is not accurate or not comprehensive enough, there will be no opportunity for self-correction afterwards. Therefore, it would be much more desirable to upgrade this static approach into an iterative and dynamic algorithm that can allow the RL agent to continuously improve its induced list of strategies and its policy model based on interactions with the environment.

For this purpose, in DYSTIL we propose to dynamically combine LLM strategy induction with on-policy reinforcement learning. Below we describe our detailed procedures in sequential order:

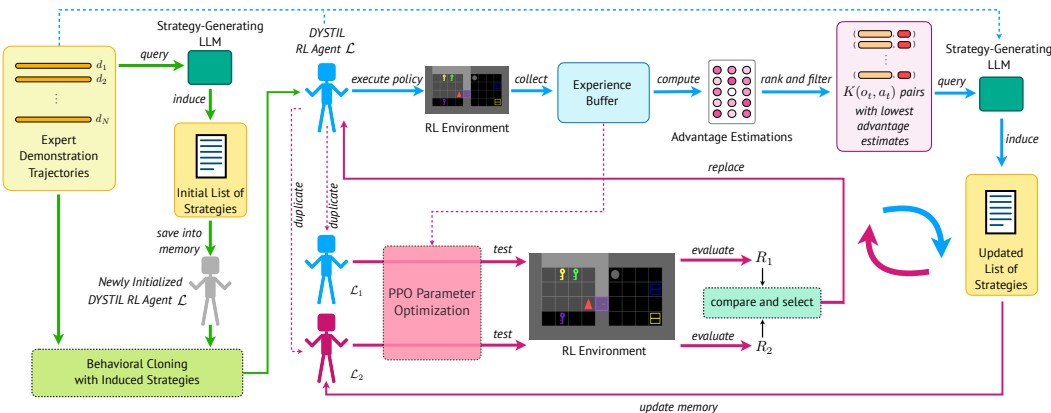

Figure 4: An overview of our proposed new modeling pipeline - Dynamic Strategy Induction of LLMs (DYSTIL) for reinforcement learning. The steps depicted in green arrows corresponds to *Initialization of the RL Agent Model*, *Initial Strategy Induction from Expert Demonstrations* and *Behavioral Cloning with Induced Strategies*; the steps depicted in blue arrows corresponds to *Experience Collection and Advantage Estimation* and *Induction of New Candidate List of Strategies*; and the steps depicted in magenta arrows corresponds to *Strategy-Integrated Proximal Policy Optimization*

**Initialization of the RL Agent Model**    To begin with, we first construct and initialize a new DYSTIL RL agent model (as introduced in Section 2.3) as our RL agent $\mathcal{L}$. In particular, we create a new empty memory $\mathcal{M}_{\mathcal{L}}$ in $\mathcal{L}$ to save its most recently updated list of strategies in real time. The parameters of the core reasoning LLM and the language modeling head of $\mathcal{L}$ are initialized from the pre-trained checkpoint of the corresponding LLM, and the parameters of the value network are randomly initialized from scratch.

**Initial Strategy Induction from Expert Demonstrations**    Now in this step, we use the method described in Section 2.2 to query a strategy-generating LLM $\mathcal{Q}$ (e.g. GPT-4o (OpenAI, 2024)) to induce an initial list of strategies $\mathcal{S}_0$ from all the expert demonstration trajectories in $\mathcal{D}$, and store $\mathcal{S}_0$ in the agent $\mathcal{L}$'s memory $\mathcal{M}_{\mathcal{L}}$. We denote the prompt template used in this step as $\mathcal{P}_{\text{initial}}$.

**Behavioral Cloning with Induced Strategies**    Next, we run behavioral cloning (Pomerleau, 1988) through supervised learning to train our RL agent model $\mathcal{L}$ to imitate the action policy in the set of expert demonstration trajectories $\mathcal{D}$. More specifically, we run optimization procedures (such as Adam (Kingma & Ba, 2015)) to gradually minimize the mean cross-entropy loss between the action distributions $\pi_{\mathcal{L}}(o \mid g, \mathcal{S}_0)$ generated by our agent model $\mathcal{L}$ and the action choices made by the expert across all the observations contained in $\mathcal{D}$, subject to a small entropy regularization (Williams & Peng, 1991; Mnih et al., 2016; Ahmed et al., 2019). Note that during behavioral cloning training we only update the parameters of the core reasoning LLM and its corresponding language modeling head, and keep the value network frozen. Intuitively, this behavioral cloning training process is very important in that it helps the agent model $\mathcal{L}$ to gradually internalize the list of induced text strategies through parameter tuning. This helps the agent model better understand how to reason with the strategies to make good action decisions under realistic scenarios in the RL environment.

**Experience Collection and Advantage Estimation**    After the RL agent model $\mathcal{L}$ has been properly trained through behavioral cloning with its initial list of strategies $\mathcal{S}_0$ over expert demonstrations $\mathcal{D}$, we follow the practice of the proximal policy optimization (PPO) algorithm (Schulman et al., 2017) to run agent $\mathcal{L}$ to execute its current policy $\pi_{\mathcal{L}}(o \mid g, \mathcal{S})$ in the RL environment $E$ for $T$ time steps to collect an experience buffer $\mathcal{B}$ containing $T$ (observation, action, reward) triples. Then, we follow the standard PPO procedures in (Schulman et al., 2017) to compute the estimated values $\hat{A}$ of the advantage function $A$ for all the $T$ (observation, action) pairs in the current experience buffer $\mathcal{B}$.

**Induction of New Candidate List of Strategies**    One important limitation of existing methods for rule induction with LLMs for sequential decision making tasks is the lack of a credit assignment mechanism that can clearly inform the LLMs which specific action decisions are mainly responsible for the eventual success or failure of different trajectories (Zhao et al., 2024), thus significantly limiting their reasoning ability to analyze how to best adjust its induced rules to correct unfavorable

action decisions. In reinforcement learning, estimation of the advantage function (Sutton et al., 1999; Schulman et al., 2016) is a commonly used technique for solving the credit assignment problem. So in DYSTIL, we use the advantage estimates calculated in the previous step to filter out the most suspiciously problematic (observation, action) pairs that could contribute the most to low episode returns, and to help the strategy-generating LLM to efficiently discern which strategy items need revision and update.

More specifically, in this step, we first rank all the $T$ (observation, action) pairs $\{(o_t, a_t)\}_{t=1}^{T}$ in the current experience buffer $\mathcal{B}$ according to their current advantage estimates $\hat{A}(o_t, a_t)$, and then filter out $K$ pairs with the lowest advantage estimates. We denote the set of these $K$ $(o, a)$ pairs as $\mathcal{H}_K$. Next, we use another prompt template $\mathcal{P}_{\text{dynamic}}$ to include textual descriptions of both $\mathcal{D}$ and $\mathcal{H}_K$ and the agent $\mathcal{L}'s$ current list of strategies $\mathcal{S}$ to query the strategy-generating LLM $\mathcal{Q}$ again to induce and generate a revised and updated list of strategies $\mathcal{S}'$. The prompt template $\mathcal{P}_{\text{dynamic}}$ that we use for this step is shown in Appendix E. Here in $\mathcal{P}_{\text{dynamic}}$ we adapt and extend the operation options in (Zhao et al., 2024) to allow the LLM $\mathcal{Q}$ to correct, add and delete existing strategy items in the list of strategies.

**Strategy-Integrated Proximal Policy Optimization**    Since in reinforcement learning the value and advantage estimations computed by the value network are not always entirely accurate, and the outputs generated by the strategy-generating LLM also have inherent randomness and noise, we should not always unconditionally trust that the newly induced list of strategies $\mathcal{S}'$ obtained from the previous step is indeed better than the current list of strategies $\mathcal{S}$. Therefore, here we adopt a propose-and-test approach - we treat $\mathcal{S}'$ only as a proposed candidate for a better strategy list, and run policy optimizations followed by empirical tests to decide whether we should replace $\mathcal{S}$ by $\mathcal{S}'$ depending on their real performance. Our detailed procedures are: (1) we make two copies of the current version of our RL agent model $\mathcal{L}$, which we denote by $\mathcal{L}_1$ and $\mathcal{L}_2$; (2) we store $\mathcal{S}$ in $\mathcal{L}_1$'s memory $\mathcal{M}_{\mathcal{L}_1}$, and replace $\mathcal{L}_2$'s memory $\mathcal{M}_{\mathcal{L}_2}$ with $\mathcal{S}'$; (3) we follow the practice of the proximal policy optimization (PPO) algorithm (Schulman et al., 2017) to update model parameters of both $\mathcal{L}_1$ and $\mathcal{L}_2$ to-

---

**Algorithm 1:** Dynamic Strategy Induction with LLMs for Reinforcement Learning (DYSTIL)

**Input:** $E, \mathcal{D}, \mathcal{Q}, \mathcal{P}_{\text{initial}}, \mathcal{P}_{\text{dynamic}}$
**Initialize:** $\mathcal{L}, \mathcal{M}_{\mathcal{L}}$
**Hyperparameters:** $T, K, N_{\text{epoch}}$
Use $\mathcal{P}_{\text{initial}}(\mathcal{D})$ to query $\mathcal{Q} \rightarrow \mathcal{S}_0, \mathcal{M}_{\mathcal{L}} \leftarrow \mathcal{S}_0$
Run `Behavioral Cloning` on $\mathcal{L}$ over $\mathcal{D}$
**for** $i = 1, 2, ..., N_{\text{epoch}}$ **do**
    Run $\mathcal{L}$ in $E$ for $T$ time steps to collect $\rightarrow$
      $\mathcal{B} = \{(o_t, a_t, r_t)\}_{t=1}^{T}$
    Compute advantages $A(o_t, a_t)$ for $t = 1$ to $T$
      using $\mathcal{L}$ and $\mathcal{B}$
    Sort $\{(o_t, a_t)\}_{t=1}^{T}$ according to $A(o_t, a_t)$
    Select the $K$ $(o_t, a_t)$ pairs from $\{(o_t, a_t)\}_{t=1}^{T}$
      with lowest $A(o_t, a_t)$ values to form a set $\mathcal{H}_K$
    Use $\mathcal{P}_{\text{dynamic}}(\mathcal{H}_K, \mathcal{M}_{\mathcal{L}}, \mathcal{D})$ to query $\mathcal{Q} \rightarrow \mathcal{S}'$
    $\mathcal{L}_1 \leftarrow \mathcal{L}, \mathcal{L}_2 \leftarrow \mathcal{L}, \mathcal{M}_{\mathcal{L}_2} \leftarrow \mathcal{S}'$
    Run `PPO-Optimization` over $\mathcal{L}_1$ w.r.t $\mathcal{B}$
    Run `PPO-Optimization` over $\mathcal{L}_2$ w.r.t $\mathcal{B}$
    Test $\mathcal{L}_1$ in $E \rightarrow R_1$; Test $\mathcal{L}_2$ in $E \rightarrow R_2$
    **if** $R_2 > R_1$ **then**
      |   $\mathcal{L} \leftarrow \mathcal{L}_2$
    **else**
      └   $\mathcal{L} \leftarrow \mathcal{L}_1$

**Return:** $\mathcal{L}$

---

wards optimizing the same standard PPO clipped surrogate objective function (Schulman et al., 2017) computed from the current experience buffer $\mathcal{B}$; (4) run empirical tests of both $\mathcal{L}_1$ and $\mathcal{L}_2$ in the RL environment $E$ to compute their respective mean average returns $R_1$ and $R_2$; (5) if $R_1 >= R_2$, then we update our agent model $\mathcal{L}$ to be $\mathcal{L}_1$ (and thus keep the same strategy list $\mathcal{S}$); if $R_2 > R_1$, then we update our agent model $\mathcal{L}$ to be $\mathcal{L}_2$ (and thus also update the agent's strategy list to be the new list $\mathcal{S}'$). Now we go back to the previous step *Induction of New Candidate List of Strategies* again.

As we can see, in DYSTIL these last two steps *Induction of New Candidate List of Strategies* and *Strategy-Integrated Proximal Policy Optimization* will be executed in cycle to iteratively train the RL agent model to improve its performance. Our DYSTIL learning framework is illustrated in Figure 4 and also summarized in Algorithm 1.

## 3 EXPERIMENTS

We evaluate the performance of DYSTIL in four challenging RL environments: the *Dynamic Obstacles* environment from the Minigrid library (Chevalier-Boisvert et al., 2023), and the *Unlock Pickup* environment, the *Key Corridor* environment and the the *Put Next* environment from the BabyAI library (Chevalier-Boisvert et al., 2019). Both Minigrid (Chevalier-Boisvert et al., 2023) and BabyAI (Chevalier-Boisvert et al., 2019) are popularly used libraries of grid-world reinforcement learning environments that are designed to have good support for language grounding. All the RL environments in Minigrid and BabyAI are partially observable in that an agent can only see a field of view of $7 \times 7$ grid cells in front of it (subject to object occlusion) at every time step. These four RL environments we use all have sparse and delayed rewards, and require complex reasoning over higher-level abstractions.

### 3.1 RL ENVIRONMENTS FOR EVALUATION

**Dynamic Obstacles**    Dynamic Obstacles is a challenging dynamic RL environment from the Minigrid library (Chevalier-Boisvert et al., 2023). In this environment, the agent's goal is to navigate through a room with moving obstacles to get to a green goal square without hitting any of them along the way (Chevalier-Boisvert et al., 2023). If the agent succeeds, it will be given a single reward of value $r = 1 - 0.9 \times (\text{total\_steps}/\text{max\_steps})$ at the final step; if it failed within maximum allowed number of steps, it will receive a reward of 0; if it hits an obstacle along the way, it will receive a $-1$ penalty reward and the episode also terminates (Chevalier-Boisvert et al., 2023). This environment is one of the most challenging ones in Minigrid because it is a dynamic and stochastic RL environment, and thus requires the agent to have strong abilities to reason about the high-level mechanisms and principles of this environment in order to make good action decisions in a safely manner. In our experiment we use the *MiniGrid-Dynamic-Obstacles-6x6-v0* configuration.

**Unlock Pickup**    Unlock Pickup is a challenging static RL environment from the BabyAI library (Chevalier-Boisvert et al., 2019). In each run of this environment, a target box is locked behind a door, and your goal as an agent is to obtain the key to unlock that door and then pick up the box using as few time steps as possible. And similar to the Dynamic Obstacles environment, the agent will receive either a single reward of value $r = 1 - 0.9 \times (\text{total\_steps}/\text{max\_steps})$ upon successful completion of the assigned task, or 0 reward if it failed within maximum allowed number of steps. Unlock Pickup is mainly difficult for its high requirement on the agent's abilities of maze exploration and navigation, avoidance of obstructions, optimal path finding, and long-horizon task planning. In our experiment we use the *BabyAI-UnlockPickupDist-v0* configuration and set max\_steps = 60.

**Key Corridor**    Key Corridor is another challenging static RL environment from the BabyAI library (Chevalier-Boisvert et al., 2019). In each run of this environment, the agent needs to explore a complex maze constituted of multiple rooms to find a key and then use that key to open a locked door in order to pick up a designated object locked behind that door, using as few time steps as possible. And again, the agent will receive either a single reward of value $r = 1 - 0.9 \times (\text{total\_steps}/\text{max\_steps})$ upon successful completion of the assigned task, or 0 reward if it failed within maximum allowed number of steps. In our experiment we use the *BabyAI-KeyCorridorS3R2-v0* configuration and set max\_steps = 60.

**Put Next**    Put Next is another challenging static RL environment from the BabyAI library (Chevalier-Boisvert et al., 2019). In each run of this environment, the agent will be assigned a randomly generated task in the form of moving a designated object to a position next to another designated object using as few time steps as possible. And similar to the Dynamic Obstacles environment, the agent will receive either a single reward of value $r = 1 - 0.9 \times (\text{total\_steps}/\text{max\_steps})$ upon successful completion of the assigned task, or 0 reward if it failed within maximum allowed number of steps. Put Next is mainly difficult for its high requirement on the agent's abilities of maze exploration and navigation, avoidance of obstructions, optimal path finding, and long-horizon task planning. In our experiment we use the *BabyAI-PutNextS5N2-v0* configuration and set max\_steps = 60.

### 3.2 OBSERVATION-TO-TEXT TRANSFORMATION

As discussed in Section 2.1, a prerequisite for performing language-grounded reinforcement learning is to have a good observation-to-text converter. In our experiments, we employ the text description

| Methods | Stra | Dynamic Obs | | Unlock Pickup | | Key Corridor | | Put Next | | Average | |
|---|---|---|---|---|---|---|---|---|---|---|---|
| | | MR | SR % | MR | SR % | MR | SR % | MR | SR % | MR | SR % |
| ReAct | ✗ | −0.014 | 51 | 0 | 0 | 0.078 | 17 | 0.143 | 24 | 0.052 | 23 |
| GLAM$_{BC}$ | ✗ | −0.747 | 13 | 0.017 | 4 | 0.210 | 40 | 0.109 | 18 | −0.103 | 18.75 |
| GLAM$_{BC+PPO}$ | ✗ | −0.688 | 16 | 0.024 | 6 | 0.204 | 37 | 0.106 | 17 | −0.088 | 19 |
| DYSTIL$_{BC}$ | ✓ | 0.096 | 47 | 0.032 | 9 | 0.259 | 46 | 0.162 | 22 | 0.137 | 31 |
| DYSTIL$_{BC+PPO}$ | ✓ | **0.248** | **65** | **0.041** | **10** | **0.280** | **56** | **0.217** | **32** | **0.197** | **40.75** |

Table 1: Our experiment results of DYSTIL and the two baseline methods ReAct Yao et al. (2023) and GLAM (Carta et al., 2023) on the Dynamic Obstacles environment from Minigrid (Chevalier-Boisvert et al., 2023), and the Unlock Pickup environment, the Key Corridor environment, and the Put Next Environment from BabyAI (Chevalier-Boisvert et al., 2019). The **Strategy** (abbreviated as Stra) column indicates whether the learning method utilizes textual strategies in its pipeline. The methods' performance scores are reported in the standard RL evaluation metrics of both *mean return* (MR) and *success rate* (SR) in percentage. For DYSTIL and GLAM we report their performance scores for two different settings: the Behavioral-Cloning-only setting (BC) and the Behavioral-Cloning-plus-PPO setting (BC+PPO). Rows showing the results of our DYSTIL methods are highlighted in light pink. The highest score in each metric is highlighted in **bold**.

generator of BabyAI-text proposed in (Carta et al., 2023) to transform an agent's raw observation in the Minigrid and BabyAI environments into a list of sentence descriptions. See Figure 7 in Appendix C for examples.

### 3.3 BASELINE METHODS

In our experiments we compare DYSTIL with two state-of-the-art baseline methods for language-grounded sequential decision making: ReAct (Yao et al., 2023) and GLAM Carta et al. (2023). GLAM can essentially be viewed as the non-strategy ablated version of DYSTIL. Here we follow GLAM Carta et al. (2023) to set $H = 2$ for all the models. For fair comparison, for GLAM we also employ the same input design and the actor module design as in DYSTIL introduced in this paper, and we take the finetuning approach for ReAct.

### 3.4 EXPERIMENT SETUP

**Model Configurations**    In our experiments, for DYSTIL we use Llama 3.1 8B Instruct (Meta, 2024), one of the best performing lightweight open-source LLMs, as the core reasoning LLM, and use GPT-4o (OpenAI, 2024), one of the SOTA closed-source large-scale LLMs, as the strategy-generating LLM. And for fair comparison, we also use Llama 3.1 8B Instruct as the decision-making LLM module for GLAM and ReAct, and we also use GPT-4o to generate thought annotations for ReAct.

**Expert Demonstrations**    We collect a set of 5 expert demonstration trajectories for each of the four RL environments.

**Training Pipelines**    For DYSTIL and GLAM, our training process consists of two stages: Behavioral Cloning (BC) and Proximal Policy Optimization (PPO). During the Behavioral Cloning stage, we use supervised learning to train the RL agent to imitate the action policy demonstrated in the set of expert trajectories for 10 epochs, and then feed the output model checkpoint into the PPO training stage and run the standard PPO algorithm for GLAM and run the DYSTIL version of PPO (as described in Section 2.4) for DYSTIL, both for 10000 training frames. Our training hyperparameters are detailed in Table 3 of Appendix D.

### 3.5 EXPERIMENT RESULTS AND ANALYSIS

**Main Results**    Our main experiment results are summarized in Table 1. As we can see from the results, DYSTIL$_{BC+PPO}$ receives the highest mean return and achieves the highest success rate for all four environments. On average DYSTIL$_{BC+PPO}$ outperforms the strongest baseline method ReAct by a significant margin of 0.145 in mean return and 17.75% in success rate in these four challenging RL environments. And notably, for the behavioral-cloning-only scores, on average DYSTIL$_{BC}$ also

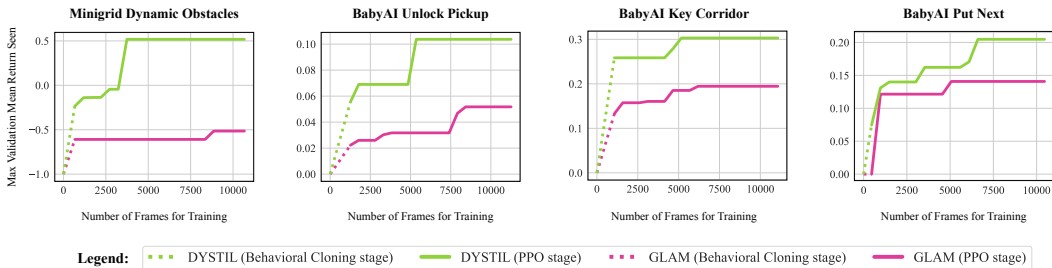

Figure 5: Comparison of sample efficiency between DYSTIL and GLAM on the four RL environments. The y-axis plots the maximum score of mean return evaluated on the validation set of environment configurations that the agent has seen so far during the learning process, and x-axis plots the number of frames of training data that has been fed into the learning pipeline so far.

outperforms GLAM$_{BC}$ by a large margin of 0.240 in mean return and 12.25% in success rate. These results demonstrate that the integration of dynamically induced textual strategies through DYSTIL can have a significant boost in the performance of both behavioral cloning and reinforcement learning paradigms.

**Sample Efficiency**     In Figure 5 we compare the sample efficiency between DYSTIL and the non-strategy baseline method GLAM. As we can see, for all four RL environments DYSTIL quickly achieves significantly higher mean return scores when consuming the same amount of training frames than GLAM across both the Behavioral Cloning stage and the PPO stage of the learning process. This empirically demonstrates that DYSTIL also enjoys higher sample efficiency than GLAM.

**Model Interpretability**     In our experiments, DYSTIL also demonstrates superior model transparency and interpretability during the reinforcement learning process. More specifically, DYSTIL provides us with a direct textual channel to observe and interpret the evolution of the implicit strategies underlying the agent's policy during reinforcement learning, which can not be achieved by previous RL methods. For example, in Appendix F, we illustrate a direct comparison between the initial list of strategies and the best list of strategies (corresponding to the highest-performing model checkpoint) acquired by the RL agent during DYSTIL training in the Dynamic Obstacles environment to show the evolution of the agent's strategies. From this comparison we can clearly see that during DYSTIL the RL agent has been dynamically improving its list of strategies by revising inaccurate items and adding new helpful strategies into the list based on its empirical interactions with the environment.

## 3.6 ABLATION STUDY

In our ablation study, we remove the dynamic strategy update component from our proposed DYSTIL procedures, and run experiments in the four RL environments to see how that will affect the performance of RL training. After the removal of the dynamic strategy update component, the RL agent will keep using the initial list of strategies that it obtains from the Strategy-Generating LLM (before behavioral cloning) for the whole PPO training process without updating it, and we call this ablated method DYSTIL$_{BC+PPO}$-Static. The results of our ablation study are listed in Table 2. As we can see from Table 2, on average the success rate drops by 6.25% after removing the dynamic strategy update component from DYSTIL, which shows that the dynamic strategy update component is indeed critical in achieving the best reinforcement learning performance with DYSTIL.

| Methods | Dynamic Obs | | Unlock Pickup | | Key Corridor | | Put Next | | Average | |
|---|---|---|---|---|---|---|---|---|---|---|
| | MR | SR % | MR | SR % | MR | SR % | MR | SR % | MR | SR % |
| DYSTIL$_{BC+PPO}$ | **0.248** | **65** | **0.041** | **10** | **0.280** | **56** | **0.217** | **32** | **0.197** | **40.75** |
| DYSTIL$_{BC+PPO}$-Static | 0.056 | 49 | 0.037 | 8 | 0.258 | 48 | 0.191 | 27 | 0.13490 | 33 |

Table 2: Ablation Study

## 4 RELATED WORK

**LLMs for Reinforcement Learning and Language-Grounded RL**    Traditionally, most policy models of deep RL algorithms have been directly operating over low-level raw features of environment observations (Mnih et al., 2013). This design choice has inevitably restricted these RL methods' abilities to learn higher-level abstractions and concepts about the RL tasks. Recently more research efforts has been made on grounding reinforcement learning into natural language (Chevalier-Boisvert et al., 2019; 2023; Carta et al., 2023; Poudel et al., 2023) and use pre-trained LLMs as the policy generator of RL agents (Carta et al., 2023). DYSTIL differs from these existing methods by enabling LLM-based RL agents to efficiently learn higher-level strategies and abstractions of the RL tasks through strategy induction from large-scale LLMs.

**LLMs for Sequential Decision Making**    Recently there has been a series of works that explore different approaches for applying LLMs to sequential decision making tasks (Yao et al., 2023; Shinn et al., 2023; Zhao et al., 2024; Yao et al., 2024). All these existing methods have two major limitations: (1) they all require querying the API of a large-scale closed-source LLM for the agent's decision making at every single time step, which make them highly infeasible for many important real-world tasks and applications that require fast inference speed to make timely decisions or require offline and lightweight deployment to integrate with operating hardware (such as robots); (2) they all rely on prompting to make inference of action decisions with frozen closed-source LLMs at every single time step, and thus do not support parametrized policy learning. In contrast, for DYSTIL the decision making inference at all time steps is run on a lightweight open-source LLM that supports full model parameter tuning. As a result, DYSTIL has the advantage of fast real-time inference during decision making, easy deployment over different application scenarios, and compatibility with on-policy reinforcement learning algorithms, while still being able to learn high-level strategies through strategy distillation from large-scale closed-source LLMs.

## 5 CONCLUSION AND BROADER IMPACT

In this paper we presented DYSTIL, a novel strategy-based neuro-symbolic reinforcement learning framework integrated with large language models. We carried out empirical experiments over challenging RL environments to evaluate DYSTIL on the task of reinforcement learning from expert demonstrations, and the results show that DYSTIL significantly outperforms state-of-the-art baseline methods while exhibiting higher sample efficiency and superior model interpretability.

**Broader Impact** Our work opens up new possibilities in leveraging powerful large language models to generate textual strategies to help reinforcement learning algorithms improve their learning performance, expedite their learning processes, and making their policy evolution more transparent. In addition, our work also demonstrates the advantages of neuro-symbolic RL methods over traditional neural-only RL methods. Last but not the least, in future works on LLM evaluation, it could also be of research interest to include new evaluation metrics and benchmarks to quantitatively measure how much performance gain can the textual strategies induced by different LLMs bring to reinforcement learning algorithms. This could serve as an interesting new aspect to gauge the knowledge reasoning and induction abilities of LLMs under cross-modal scenarios.

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

# A  EXAMPLE MODEL INPUT

---

**Model Input**

**Possible action of the agent:** *left turn, right turn, move forward.*

**Rules to follow:**

**1. Advance when clear and safe:**
  - The agent should move forward when there is no immediate obstacle (e.g., no ball directly in front or very close on either side) and when the green goal is within a reasonable distance (1-3 steps away). This minimizes time steps without unnecessary rotations.

$$\vdots$$

**10. Move forward when the goal is directly ahead, even if other obstacles are nearby:**
  - If the green goal is directly ahead and no obstacles block the forward path, the agent should prioritize moving forward towards the goal, regardless of surrounding obstacles, as the task can be completed in fewer steps.

**Goal of the Agent:** get to the green goal square
**Observation 1:** you see a wall 1 step left, you see a blue ball 1 step forward, you see a blue ball 2 steps right, you see a green goal 3 steps right and 2 steps forward, you see a blue ball 3 steps right.
**Action 1:** right turn.
**Observation 2:** you see a wall 4 steps forward, you see a wall 3 steps left, you see a wall 2 steps right, you see a green goal 2 steps left and 3 steps forward, you see a blue ball 2 steps left and 1 step forward, you see a blue ball 1 step left and 3 steps forward, you see a blue ball 1 step left and 2 steps forward.
**Action 2:** move forward.
**Observation 3:** you see a wall 3 steps left, you see a wall 2 steps right, you see a green goal 2 steps left and 2 steps forward, you see a blue ball 2 steps forward, you see a blue ball 1 step forward.
**Action 3:**

---

Figure 6: An example textual input into our proposed Strategy-Integrated LLM Actor-Critic Model for $H = 2$. This example input is constructed when the RL agent is traversing the Dynamic Obstacles RL environment from the Minigrid library (Chevalier-Boisvert et al., 2023).

# B  DESIGN DETAILS OF THE ARCHITECTURE OF THE RL AGENT'S ACTOR MODULE IN DYSTIL

In coordination with the rise of research interests in language-grounded RL, recent works have also been exploring the direction of using language models as the core policy generators of the agents in reinforcement learning. For example, (Carta et al., 2023) proposes an architecture for a policy LLM that directly takes a textual prompt comprised of an environment introduction, a task description, a historical trajectory of observation descriptions and action names, and an action prompting phrase as input, and then feed this prompt into an encoder-decoder language model to output the conditional probability of each token in each action name given the prompt and the generated action tokens through its language modeling head plus softmax. It then multiplies such condition probabilities for all the tokens in each action name together, and then normalize to obtain a probability distribution over the set of all possible actions to serve as its policy (Carta et al., 2023). This architecture suffers a lot from the issue of slow inference, because for generating each single action decision we need to run this policy LLM for $N_A \times M_A$ times (Carta et al., 2023), where $N_A$ is the total number of possible actions and $M_A$ is the average number of tokens in all the action names. In this work, in order to improve inference speed and training efficiency, we design an upgraded architecture for the output side of the LLM policy generator (i.e. the actor module of our DYSTIL agent). First, if necessary, we make some small tweaks on the names of the actions such that no two actions would share the same first token in their names (e.g. we could change the two action names '*turn left*' and '*turn right*' into '*left turn*' and '*right turn*' to avoid first token conflict). Next, when generating an action decision we only need to run the policy LLM once and then we can directly take the logits corresponding to the first token of each action name outputed by the language modeling head, group them into a vector, and then run softmax to obtain a proability distribtion over the set of all possible actions as our policy.

## C  EXAMPLES OF OBSERVATION-TO-TEXT TRANSFORMATION

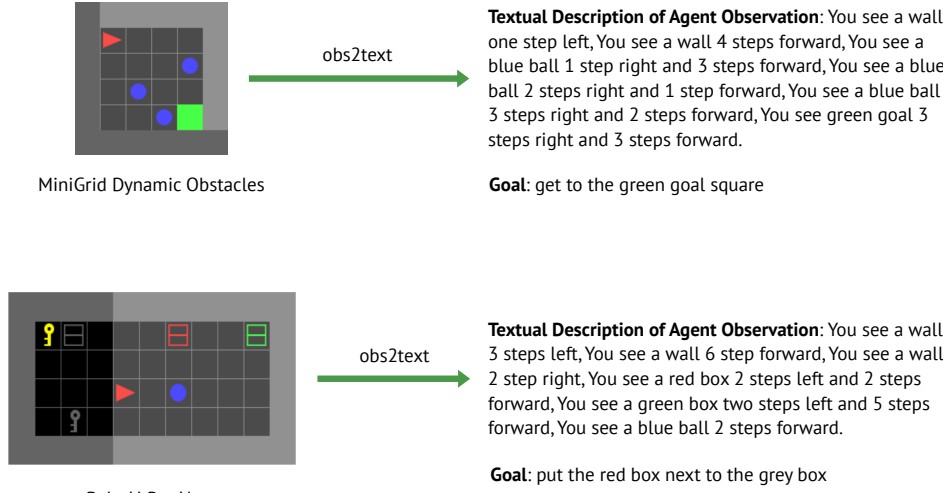

Figure 7: Examples of Observation-to-Text Transformation in Minigrid and BabyAI environments using the text description generator of BabyAI-text proposed in (Carta et al., 2023).

## D  TRAINING HYPERPARAMETER SETTINGS

Table 3: Training Hyperparameter Settings

| Hyperparameter | Value |
|---|---|
| Behavioral Cloning Hyperparameters | |
| Batch size | 16 |
| Learning rate | $1 \times 10^{-4}$ |
| PPO Hyperparameters | |
| Batch size | 32 |
| Learning rate | $1 \times 10^{-5}$ |
| Number of processes | 4 |
| Number of frames per processes between updates | 128 |
| GAE $\lambda$ | 0.95 |
| Entropy coefficient | 0.01 |
| Value coefficient | 0.5 |
| DYSTIL Hyperparameters | |
| Hidden size of the critic network | 1024 |
| Number of $(o, a)$ pairs for new strategy induction | 10 |

# E   PROMPT TEMPLATE $\mathcal{P}_{\text{DYNAMIC}}$

Imagine now you are a reinforcement learning agent in a 2D gridworld RL platform called MiniGrid, and you are learning to complete tasks in a specific RL environment called 'Dynamic Obstacles' on this Minigrid platform. This 'Dynamic Obstacles' environment is an empty room with moving obstacles. In each run of this 'Dynamic Obstacles' task in this RL environment, your goal as an agent is to reach the green goal square using as few time steps as possible without colliding with any obstacle. If the agent collides with an obstacle, a large penalty is subtracted and the episode is terminated. Your possible actions as an agent at each time step are: 'left turn', 'right turn', and 'move forward'.

You are provided with 5 successful trajectories of expert demonstrations of the oracle courses of actions to complete tasks in this 'Dynamic Obstacles' environment for your reference, which are listed in detail below:

. . .

Currently, as the reinforcement learning agent, you are following the following list of strategies when making action decisions in this 'Dynamic Obstacles' environment:

. . .

And in your current iteration of experience collection during a PPO training process, the following 10 state-action pairs (they may come from different episodes) received the lowest advantage values, which indicates that these action decisions might not be optimal:

. . .

Now upon analyzing the above 10 state-action pairs with low advantage values, and based on your analysis and understanding of the 5 expert demonstrations of oracle trajectories provided to you earlier, please modify and update the list of strategies that you are currently following if you are confident that it is appropriate to do so. You can correct existing strategy items if you think they are inaccurate, you can add new strategy items if you think they are currently missing, and you can delete existing strategy items if you think they are wrong. Please remember that the above advantage values are estimated by the value network of the RL agent model during PPO training, and thus may not be entirely accurate and should be analyzed with caution. Therefore, you should consider the evidence suggested by the above observation-action pairs with low advantage values, the patterns and insights exhibited by the expert demonstration trajectories, and your own understanding, reasoning and judgement about this 'Dynamic Obstacles' task all together to make wise decisions when modifying and updating the list of strategies. Please only return the updated list of strategies without any other text before or after the list.

# F    EXAMPLE OF STRATEGY EVOLVEMENT

**The initial list of strategies learned by the DYSTIL RL agent in the Minigrid Dynamic Obstacles environment:**

1. Advance when clear and safe:

- The agent should move forward when there is no immediate obstacle (e.g., no ball directly in front or very close on either side) and when the green goal is within a reasonable distance (1-3 steps away). This minimizes time steps without unnecessary rotations.

2. Turn towards the goal when necessary:

- The agent should make a left or right turn to align with the direction of the green goal when moving forward directly is blocked by obstacles or walls. Turning should be based on the goal's relative position to the agent.

3. Avoid obstacles by turning instead of moving forward:

- The agent should prioritize turning when it detects obstacles (e.g., blue balls) directly in front. Moving forward into an obstacle should always be avoided as it ends the episode with a penalty.

4. Maintain goal orientation after turning:

- After turning, the agent should quickly readjust and move towards the green goal if the path is clear, ensuring minimal time is spent making unnecessary turns.

5. Turn early when a wall is close:

- The agent should turn left or right early if a wall is detected within 1-2 steps ahead. Moving forward into a wall wastes a time step, and adjusting earlier is more efficient.

6. Favor forward movement when obstacles are distant:

- When obstacles are visible but farther than 1-2 steps away, the agent should continue moving forward as long as the path is clear. Turning preemptively when obstacles are distant can delay the agent unnecessarily.

7. Turn towards the side of the goal if obstacles are scattered ahead:

- If multiple obstacles (blue balls) are scattered ahead and there's no clear direct path to the green goal, the agent should turn towards the side where the green goal is closer and avoid the cluster of obstacles.

8. Optimize by following the closest safe route to the goal:

- When the green goal is identified within 2-3 steps but obstacles obstruct direct movement, the agent should prioritize a minimal deviation (either left or right turn) to navigate around the obstacles and then move towards the goal.

9. Minimize unnecessary rotations:

- The agent should avoid making consecutive left-right turns in short succession unless absolutely necessary for obstacle avoidance. The strategy should be to realign with the goal and proceed forward as much as possible.

10. Move forward when the goal is directly ahead, even if other obstacles are nearby:

- If the green goal is directly ahead and no obstacles block the forward path, the agent should prioritize moving forward towards the goal, regardless of surrounding obstacles, as the task can be completed in fewer steps.

**The best list of strategies learned by the DYSTIL RL agent in the Minigrid Dynamic Obstacles environment.**

1. Advance when clear and safe:

- The agent should move forward when there is no immediate obstacle (e.g., no ball directly in front or very close on either side) and when the green goal is within a reasonable distance (1-3 steps away). This minimizes time steps without unnecessary rotations.

2. Turn towards the goal when necessary:

- The agent should make a left or right turn to align with the direction of the green goal when moving forward directly is blocked by obstacles or walls. Turning should be based on the goal's relative position to the agent.

3. Avoid obstacles by turning instead of moving forward:

- The agent should prioritize turning when it detects obstacles (e.g., blue balls) directly in front. Moving forward into an obstacle should always be avoided as it ends the episode with a penalty.

4. Maintain goal orientation after turning:

- After turning, the agent should quickly readjust and move towards the green goal if the path is clear, ensuring minimal time is spent making unnecessary turns.

5. Turn early when a wall is close:

- The agent should turn left or right early if a wall is detected within 1-2 steps ahead. Moving forward into a wall wastes a time step, and adjusting earlier is more efficient.

6. Favor forward movement while alert to nearby obstacles:

- When obstacles are visible but not immediately in front (more than 1 step away), the agent should favor moving forward with heightened alertness, evaluating if the path remains clear or if a turn is needed to avoid upcoming obstacles.

7. Plan moves with goal proximity in mind:

- When the green goal is identified within 2-3 steps but obstacles obstruct direct movement, the agent should consider minimal deviations (left or right turns) to navigate around obstacles, ensuring quick progression towards the goal.

8. Minimize unnecessary rotations:

- The agent should avoid making consecutive left-right turns in short succession unless absolutely necessary for obstacle avoidance. The strategy should be to realign with the goal and proceed forward as much as possible.

9. Move forward when the goal is directly ahead, even if other obstacles are nearby:

- If the green goal is directly ahead and no immediate obstacles block the forward path, the agent should prioritize moving forward towards the goal, as this can be achieved in fewer steps.

10. Blend observations with historical context:

- The agent should sometimes reconsider its immediate action decision based on recent observations and actions to prevent repeated unoptimized movements (e.g., moving forward into known problematic areas).

11. Execute small direction adjustments when multiple obstacles:

- If there are multiple scattered obstacles (blue balls) ahead, the agent should make small directional adjustments (left or right turns) to better navigate through or around them while maintaining a path towards the goal.

12. Avoid repetitive turning patterns in short sequence:

- The agent should avoid alternating between left and right turns in quick succession, as this indicates a lack of efficient navigation and situational awareness, leading to suboptimal trajectories.

13. Focus on incremental progress towards the goal:

- The agent should break down the path to the goal into a series of small, manageable movements, constantly recalibrating based on the updated observation to ensure consistent progress without unnecessary detours.

14. Efficiently navigate around immediate obstacles:

- When an obstacle is detected immediately ahead (1 step), the agent should prioritize making a small directional adjustment to avoid a direct collision, while promptly reorienting towards the goal thereafter.

