# OpenReview forum: "DYSTIL: Dynamic Strategy Induction with Large Language Models for Reinforcement Learning"
_ICLR.cc/2025/Conference — Submitted to ICLR 2025_

### Official Review · Reviewer_QBcw · 2024-11-01

**Soundness:** 2
**Presentation:** 3
**Contribution:** 2
**Rating:** 5
**Confidence:** 4

**Summary:**

The paper introduces DYSTIL, a neuro-symbolic reinforcement learning framework that integrates large language models (LLMs) to dynamically induce and internalize textual strategies, enhancing policy generalization and sample efficiency. By leveraging LLMs to provide strategy guidance based on expert demonstrations, DYSTIL improves interpretability and transparency in reinforcement learning tasks. Empirical results demonstrate that DYSTIL outperforms state-of-the-art methods by 17.75% in success rate across complex environments like Minigrid and BabyAI.

**Strengths:**

- Quality: The paper presents ideas through clear text and figures, aiding understanding of the overall concepts.
- Significance: This paper demonstrates that the proposed method outperforms two baselines in grid-world environments.

**Weaknesses:**

- Scalability: DYSTIL’s reliance on closed-source, SOTA LLMs (e.g., GPT-4o) raises issues of scalability, reproducibility, and accessibility, especially for the model which needs to recurrently call strategy-generating LLM for each iteration. The paper also lacks ablation studies using different LLMs, which would help clarify the flexibility of using other LLMs for this work.

**Questions:**

1. Can DYSTIL generalize to other language-based decision-making tasks, such as those solved by ReAct (e.g., ALFWorld)? How could you extend your framework to accommodate these tasks?
2. In the GLAM baseline paper[1], the average success rate converges and reaches high performance (i.e., over 80%) at approximately 1e6 steps. Is there a reason you chose 1e5 steps for evaluation? What causes the discrepancy between your configuration and results compared to theirs?
3. In the ablation study, dynamic strategy updates are removed, so there is no $\mathcal{L}_2$ in the static strategy settings. Does this result in more iterations compared to the proposed method based on the same training frames? I also want to confirm whether $\mathcal{L}, \mathcal{L}_1, \mathcal{L}_2$'s executions are all counted in training frames.
4. Can the strategies generalize to novel tasks? For instance, would training on Unlock Pickup help in solving Key Corridor?

[1] Carta et. al. "Grounding large language models in interactive environments with online reinforcement learning". In Proceedings of ICLR 2023.

---

> ### Author Response · Authors · 2024-11-29
> **Response to Reviewer QBcw**
>
> Thank you very much for your feedback and suggestions in the review! Below are our responses to your review:
>
> ---
>
> > - Scalability: DYSTIL’s reliance on closed-source, SOTA LLMs (e.g., GPT-4o) raises issues of scalability, reproducibility, and accessibility, especially for the model which needs to recurrently call strategy-generating LLM for each iteration.
>
>
> Nowadays there are many papers that propose new methods that rely on recurrently calling and querying closed-source and SOTA LLMs (such as [R1]), which is a perfectly fine and legitimate approach that is very widely accepted and adopted in today’s AI research community. Many of these papers’ proposed methods even rely on querying closed-source and SOTA LLMs for making decisions at every single time step, while in contrast our DYSTIL method only requires querying a closed-source and SOTA LLM once every epoch. Therefore, DYSTIL is already performing much fewer callings of closed-source and SOTA LLMs than many existing works in the domain.
>
> ---
> > 1. Can DYSTIL generalize to other language-based decision-making tasks, such as those solved by ReAct (e.g., ALFWorld)?
>
> DYSTIL is designed as a reinforcement learning framework, so it should be applied to reinforcement learning tasks.
>
> ---
> > 2. In the GLAM baseline paper[1], the average success rate converges and reaches high performance (i.e., over 80%) at approximately 1e6 steps. Is there a reason you chose 1e5 steps for evaluation? What causes the discrepancy between your configuration and results compared to theirs?
>
> The reason why our RL training takes fewer training time steps than the original RL training in the GLAM baseline paper is because in our experiment we have access to a set of expert demonstration trajectories. This setup is because, as described in the Problem Formulation paragraph of Section 2.1, in this paper we target the problem of ‘reinforcement learning from expert demonstration’. Therefore, the RL algorithms are not trained from scratch, but are initialized by a checkpoint that we obtain from first running behavioral cloning over the set of expert demonstration trajectories. That’s why the following RL training is much more efficient and takes fewer training time steps than that in the GLAM baseline paper.
>
> ---
> > 3. In the ablation study, dynamic strategy updates are removed, so there is no $\mathcal{L}_2$ in the static strategy settings. Does this result in more iterations compared to the proposed method based on the same training frames? I also want to confirm whether $\mathcal{L}$, $\mathcal{L}_1$, $\mathcal{L}_2$'s executions are all counted in training frames.
>
> In the counting of training frames, we only count $\mathcal{L}$’s policy execution trajectories as these frames are the actual frames that are collected into the experience buffer and used as training data to update model parameters during each training epoch. Only these frames executed by $\mathcal{L}$ are meaningful to be counted when comparing sample efficiency of RL algorithms. In the ablation study, after the dynamic strategy updates are removed, there are actually less computation steps in each epoch as there will be no $\mathcal{L}_2$, and thus no test-evaluate-compare-select procedure in each epoch for this ablated method.
>
> ---
> > 4. Can the strategies generalize to novel tasks? For instance, would training on Unlock Pickup help in solving Key Corridor?
>
> Currently our DYSTIL framework is designed to be task-specific, as different RL tasks often tend to require different strategies. But your suggestion is very nice and constructive, and in future work we can explore the possibilities of extending our method to also enable inter-task strategy generalization to a certain extent.
>
> ---
> **References**
>
> - [R1] Yao et al. Retroformer: Retrospective large language agents with policy gradient optimization. In The Twelfth International Conference on Learning Representations, 2024

---

### Official Review · Reviewer_i2kJ · 2024-11-04

**Soundness:** 2
**Presentation:** 3
**Contribution:** 2
**Rating:** 3
**Confidence:** 4

**Summary:**

This paper proposed DYSTIL which integrates LLMs into a strategy-based neuro-symbolic reinforcement learning framework. The method aims to address the generalization issue of RL.

**Strengths:**

1. The paper is well-written and easy to follow.
2. The authors provide illustrations of their method, which makes it clear to how it works.

**Weaknesses:**

1. My biggest concern is the limited novelty and experiments. There are many papers that proposed strategy-generating methods as a summary or reflection of the past trajectories, such as [1]. The authors failed to discuss the similarities and differences between their method and these works.
2. The experiments are only conducted in several environments from Minigrid. Whether this approach can generalize and how to design each component for different tasks remains unclear. Besides, the compared baselines are limited. I strongly encourage the authors to do literature reviews and add more baselines such as [1].

[1] Shinn et al., Reflexion: Language Agents with Verbal Reinforcement Learning.

**Questions:**

See weakness.

---

> ### Author Response · Authors · 2024-11-29
> **Response to Reviewer i2kJ**
>
> Thank you very much for your feedback and suggestions in the review! Below are our responses to your review:
>
> ---
> > My biggest concern is the limited novelty and experiments. There are many papers that proposed strategy-generating methods as a summary or reflection of the past trajectories, such as [1]. The authors failed to discuss the similarities and differences between their method and these works.
>
> In fact, we have already clearly discussed the similarities and differences between our proposed DYSTIL method and Reflexion [1] as well as other works in applying LLMs to sequential decision making in Line 497-509 of our paper manuscript:
>
> > "Recently there has been a series of works that explore different approaches for applying LLMs to sequential decision making tasks (Yao et al., 2023; Shinn et al., 2023; Zhao et al., 2024; Yao et al., 2024). All these existing methods have two major limitations: (1) they all require querying the API of a large-scale closed-source LLM for the agent’s decision making at every single time step, which make them highly infeasible for many important real-world tasks and applications that require fast inference speed to make timely decisions or require offline and lightweight deployment to integrate with operating hardware (such as robots); (2) they all rely on prompting to make inference of action decisions with frozen closed-source LLMs at every single time step, and thus do not support parametrized policy learning. In contrast, for DYSTIL the decision making inference at all time steps is run on a lightweight open-source LLM that supports full model parameter tuning. As a result, DYSTIL has the advantage of fast real-time inference during decision making, easy deployment over different application scenarios, and compatibility with on-policy reinforcement learning algorithms, while still being able to learn high-level strategies through strategy distillation from large-scale closed-source LLMs."
>
> ---
> > Whether this approach can generalize and how to design each component for different tasks remains unclear.
>
> All the steps and procedures of our proposed DYSTIL method are clearly described in Section 2 of the paper in a very general manner without restricting them to any particular RL tasks or environments. Therefore, our proposed DYSTIL method is widely applicable to different RL tasks. In its design it was not specifically tailored to any particular RL tasks. For different RL tasks, you should follow the same procedures and principles as detailed in the paper to design each component of DYSTIL.

---

### Official Review · Reviewer_eNLJ · 2024-11-04

**Soundness:** 1
**Presentation:** 2
**Contribution:** 2
**Rating:** 3
**Confidence:** 4

**Summary:**

The paper proposes a way to leverage expert data for behavior cloning and RL to craft policies that are conditioned on a set of strategies which hopefully encode generalizable behavior. The authors claim that existing methods on BC+RL suffer from important issues (poor generalization, sample efficiency and interpretability), which the proposed approach can address. In particular, the authors train an open source LLM on expert data through behavior cloning by conditioning the policy on strategies that are devised by a teacher LLM (GPT-4o). The model is then trained with RL data, leading to a new list of strategies, which is then used to further guide the agent. The strategies are selected by verifying whether they help the model achieve higher performance. On a set of four environments, the paper shows that the proposed approach improves upon previous baselines.

**Strengths:**

The paper identifies an important area for research, that is, how to combine expert data and reinforcement learning. The proposed approach is different from some of the more traditional ways of leveraging both kinds of data, building on the strengths of LLMs to devise high level strategies.

**Weaknesses:**

The empirical setup brings a lot of questions. A major red flag is that there are no error bars at all and no mention of the number of seeds. Please see the rich literature on the subject of statistical relevance in decision making [1, 2].  For the tasks themselves, it is not clear why some choices are made. For example, is the max_steps=60 the default number? In the codebase of Minigrid I can see that the default value is set of 100, so an explanation would be necessary.

Another important area of doubt is concerning the strategy for updating the list of strategies. Currently, this is a complicated method that relies on evolving the strategy list with respect to performance. Why is such a complex method used? How sensitive is it to the different hyperparameters? How does it compare to simply asking GPT-4o for a new list of strategies? These are key questions that are completely unanswered.

The authors claim that generalization is a limitation of BC+RL, yet the paper does not show any experiments on generalization. This would be a great opportunity for the authors to show the compositionally that is afforded by language. It would also be a great opportunity to address another important are of concern: how much does the list of strategies affect the model? How much can you steer its behavior by changing the list? At the moment, it really isn't clear that the RL agent really responds to this conditioning.

The performance numbers reported for some of these tasks seems very low, which also comes from a limited set of baselines. In particular, I would really like to see the performance of GPT-4o on these tasks. Another family of baselines would be to compare to LLMs generating goals for the RL agent [3], which is relatively close to the ideas presented here. Notice that in that paper the results are significantly better than the numbers presented here.

**Questions:**

In the introduction, it is mentioned that BC+RL can't enable an RL agent tot acquire higher level abstractions and understanding of the RL task. This is not only very loosely defined, but likely not true. Do the authors mean that an RL agent wouldn't acquire an understanding that can be translated in language? This is very different than the claims being made.

Why use GPT-4o for generating strategies? How does Llama 3.1 compare? It would be much more preferable to have it be the same family of models.

The word "neuro-symbolic" is used to characterize the method, but is it really a neuro-symbolic method? To me it just seems like the neural network is additionally conditioned on language. This qualifier seems a bit of stretch.

[1] Deep Reinforcement Learning at the Edge of the Statistical Precipice, Agrawal et al., 2022

[2] Deep reinforcement learning that matter, Henderson et al., 2018

[3] Code as Reward: Empowering Reinforcement Learning with VLMs, Venuto et al., 2024

---

> ### Author Response · Authors · 2024-11-29
> **Response to Reviewer eNLJ - Part 1**
>
> Thank you very much for your feedback and suggestions in the review! Below are our responses to your review:
>
> ---
> > For the tasks themselves, it is not clear why some choices are made. For example, is the max_steps=60 the default number?
>
> Both Minigrid and BabyAI are highly flexible and modular RL env libraries that support researchers to customize RL testing environments that they see most fit to their specific research goals and purposes in their projects. Therefore, Minigrid and BabyAI allows researchers to set max_steps to any integer they want. Essentially, this max_steps parameter is used to control the relative difficulty of each task, and the numerical scores of an agent’s performance will also be affected as reward value is computed as ‘1 − 0.9 × (total_steps/max_steps)’. In our experiment we choose to set max_steps = 60 for ‘Unlock Pickup’, ‘Key Corridor’, ‘Put Next’ in order to: (1) increase the difficulty of these tasks so that our evaluation metrics (especially the success rate) can be better used to gauge the reasoning and planning abilities of agents; (2) speed up the RL training process.
>
> ---
> > Another important area of doubt is concerning the strategy for updating the list of strategies. Currently, this is a complicated method that relies on evolving the strategy list with respect to performance. Why is such a complex method used?
>
>
> We don't agree that the strategy update method of DYSTIL is 'complex' or 'complicated'. On the contrary, we have already followed a minimalistic approach when designing the strategy-updating procedures in DYSTIL, and all the steps are necessary and serving very important purposes as explained in our paper. In fact, the reason why we design to use advantage estimates to help guide the strategy-generating LLM to generate new list of strategies instead of simply ‘asking GPT-4o for a new list of strategies’ has already been clearly explained in Line 266-275 of the paper manuscript:
>
> > "One important limitation of existing methods for rule induction with LLMs for sequential decision making tasks is the lack of a credit assignment mechanism that can clearly inform the LLMs which specific action decisions are mainly responsible for the eventual success or failure of different trajectories (Zhao et al., 2024), thus significantly limiting their reasoning ability to analyze how to best adjust its induced rules to correct unfavorable action decisions. In reinforcement learning, estimation of the advantage function (Sutton et al., 1999; Schulman et al., 2016) is a commonly used technique for solving the credit assignment problem. So in DYSTIL, we use the advantage estimates calculated in the previous step to filter out the most suspiciously problematic (observation, action) pairs that could contribute the most to low episode returns, and to help the strategy-generating LLM to efficiently discern which strategy items need revision and update."
>
> In contrast, simply ‘asking GPT-4o for a new list of strategies’ would be aimless and thus highly inefficient and completely relying on luck.
>
> ---
> > It would also be a great opportunity to address another important are of concern: how much does the list of strategies affect the model? How much can you steer its behavior by changing the list? At the moment, it really isn't clear that the RL agent really responds to this conditioning.
>
> According to our design of the DYSTIL framework, we do not expect a new list of strategies newly induced by the strategy-generating LLM to be immediately reflected in the RL agent’s policy and behavior. Instead, according to our design of DYSTIL, any newly induced list of strategies will need to be gradually learned and internalized by the RL agent in the PPO optimization steps that follow the injection of the new list of strategies in order to be mastered by the RL agent.
>
> ---
> > The performance numbers reported for some of these tasks seems very low
>
> The numerical values of the performance numbers reported for some tasks seems ‘low’ because: (1) these tasks are by design inherently more difficult than other simpler tasks in the libraries; (2) we purposefully set ‘max_steps = 60’ as explained earlier, and thus the performance numbers’ numerical values will naturally be lower according to the way the reward is calculated in Minigrid and BabyAI environments, which is very reasonable and understandable. Also, in our experiment results, what really matters is the relative comparison of the performance numbers of different methods, not their absolute numerical values.

---

> ### Author Response · Authors · 2024-11-29
> **Response to Reviewer eNLJ - Part 2**
>
> ---
> > The word "neuro-symbolic" is used to characterize the method, but is it really a neuro-symbolic method?
>
> Yes, our DYSTIL method is indeed a very typical neuro-symbolic method. As illustrated in Figure 1 of the paper, our proposed DYSTIL RL agent has both (1) neural components, which includes its core reasoning LLM, language modeling head, and value network; and (2) a symbolic component, which is the list of strategies in the form of natural language texts. During the reinforcement learning process, DYSTIL repeatedly alternates between (1) performing explicit reasoning over symbolic rules (i.e. the list of strategies) given newly collected empirical evidence from the environment to try to improve them, and (2) performing parametrized policy optimization on the neural components of the DYSTIL RL agent in order to internalize updated strategies (symbolic rules). This makes DYSTIL a very typical neuro-symbolic method, and perfectly justifies its characterization as a neuro-symbolic method.

---

### Meta-Review · Area_Chair_uQ8x · 2024-12-21

**Metareview:**

The paper presents a neuro-symbolic reinforcement learning framework that integrates LLMs to generate and update strategies for RL agents. The framework aims to improve policy generalization and sample efficiency while maintaining interpretability. Dispite of the interesting integration of LLMs with RL, reviewers raised concerns on the limited novelty compared to existing strategy-generating methods, the scalability concerns with closed-source LLM dependency, the limited experimental scope, and the insufficient statistical validation. The authors are encouraged to improve this work from these aspects.

**Additional Comments On Reviewer Discussion:**

The discussion revealed a gap between the authors' view of their contribution and the reviewers' assessment, with limited productive dialogue after the initial rebuttals.

---

### Decision · Program_Chairs · 2025-01-22

Reject